# Induction of flight via midbrain projections to the cuneiform nucleus

**Emmy Tsang**[1☯], **Camilla Orlandini**[1,2☯], **Rahul Sureka**[1], **Alvaro H. Crevenna**[1],
**Emerald Perlas**[1], **Izzie Prankerd**[1,3], **Maria E. Masferrer**[1], **Cornelius T. Gross**[1]*

**1** Epigenetics & Neurobiology Unit, EMBL Rome, European Molecular Biology Laboratory, Monterotondo, Rome, Italy, **2** Neurobiology Master's Program, Sapienza University, Piazzale Aldo Moro, Rome, Italy, **3** University of Bath, Bath, United Kingdom

☯ These authors contributed equally to this work.
* gross@embl.it

**Data Availability Statement:** All relevant data are within the manuscript and its Supporting Information files.

## Abstract

The dorsal periaqueductal gray is a midbrain structure implicated in the control of defensive behaviors and the processing of painful stimuli. Electrical stimulation or optogenetic activation of excitatory neurons in dorsal periaqueductal gray results in freezing or flight behavior at low and high intensity, respectively. However, the output structures that mediate these defensive behaviors remain unconfirmed. Here we carried out a targeted classification of neuron types in dorsal periaqueductal gray using multiplex *in situ* sequencing and then applied cell-type and projection-specific optogenetic stimulation to identify projections from dorsal periaqueductal grey to the cuneiform nucleus that promoted goal-directed flight behavior. These data confirmed that descending outputs from dorsal periaqueductal gray serve as a trigger for directed escape behavior.

## Introduction

The periaqueductal gray (PAG) is a brainstem structure that plays a core role in the production of defensive behavior and related autonomic output as well as the processing of painful stimuli [1–8]. PAG is divided into functional domains–called columns [9]–based on histochemical stains, afferent and efferent connectivity, and electrical and chemical stimulation studies in laboratory rodents. Dorsal PAG (dPAG) consists of the dorsomedial, dorsolateral, and lateral PAG and receives the vast majority of its inputs from the thalamus and hypothalamus [5, 10–15] and provides outputs to medulla and spinal cord. Electrical stimulation of dPAG evokes freezing and flight behavior as well as sympathetic arousal in rodents, cats, and primates [16–19] and intense sensations of fear and being chased in humans [20, 21]. Cytotoxic lesions or pharmacogenetic inhibition of dPAG, on the other hand, reduce defensive responses to predator or conspecific threats [6, 22, 23] demonstrating that dPAG is a major conduit for the production of innate defensive responses to threat across mammals [24].

Recent single unit recording studies have identified at least two neural populations in dPAG that are active during approach-avoidance behavior [25–27]. One of these, called *Assessment* cells, showed a gradual increase in firing activity as the animal approached a threat, while the other, called *Flight* cells, did not respond appreciably during approach. When the animal

**Funding:** The work was supported by EMBL (https://www.embl.org/) and the European Research Council (ERC) Advanced Grant (https://erc.europa.eu/funding/advanced-grants) COREFEAR to C.T.G. and by a Croucher Foundation scholarship to E.T. The funders had no role in study design, data collection and analysis, decision to publish, or preparation of the manuscript. There was no additional external funding received for this study.

**Competing interests:** The authors have declared that no competing interests exist.

reached the closest point of approach to the threat–just before the animal turned to flee–*Assessment* and *Flight* cells showed a dramatic switch in firing activity. *Assessment* cells abruptly shut off their activity while *Flight* cells abruptly increased their firing. As the animal completed its flight, *Flight* cell activity dropped back to baseline. *Flight* cells have also been recorded during visually elicited flight to a looming stimulus where approach does not occur [22]. This dual encoding of sensory and motor behavior in separate neuron classes in dPAG is consistent with a sensory-motor transformation and its role as a trigger for defensive flight.

How does activity in dPAG trigger high speed escape behavior? Anatomical tract tracing suggested that dPAG neurons project to several downstream brainstem structures known to produce locomotor behavior, including the mesencephalic locomotor region (MLR) and lateral paragigantocellularis nucleus (LPGi) [28]. Recent cell-type specific optogenetic activation showed that a prominent target of dPAG in MLR, the cuneiform nucleus (CnF), is capable of producing high speed locomotor responses, raising the possibility that dPAG projections to CnF may provide a key output for defensive escape [29–32]. Here we use multiplex spatial gene expression analysis to identify a core set of neuronal cell types in dPAG, functionally test whether they are able to mediate defensive escape, examine their anatomical projections to CnF, and determine whether these projections are able to drive goal-directed flight behavior in laboratory mice. Our findings argue that excitatory dPAG projections to CnF are likely to be a major dPAG output for high-speed defensive escape.

## Results

### Single-cell multiplex gene expression profiling identifies dorsal PAG cell-types

Initially, we carried out single cell multiplex gene expression profiling to classify neuron types in the dorsal periaqueductal gray (dPAG) of the mouse brainstem for subsequent functional characterization. Multiplex *in situ* sequencing (ISS) [33–35] was used to localize a set of known marker genes (*Vglut2*, *Vgat*, *Gad2*, *Nos1*, *Map2*, *Grin2b*, and *Tac2*) in fresh frozen brain sections of PAG taken just rostral to the oculomotor nucleus. The spatial distribution of transcripts matched well to those reported previously for the individual genes with enrichment of *Gad2* and *Vgat* in vlPAG, *Nos1* in dlPAG, and *Tac2* in dmPAG (Allen Brain Atlas; **Fig 1A**, **S2 Fig**). Co-localization to putative cell bodies in dPAG using a DAPI-derived selective spatial mask and stringent quality control signal thresholding revealed clearly segregating populations of excitatory and inhibitory neurons (*Vglut2+/Gad2+/Vgat+* vs. *Vglut2+*, 21/2183; *Vglut2+/Gad2+/Vgat+* vs. *Vgat+/Gad2+*, 181/1076; **Fig 1B**). The relatively stringent quality control for gene transcript detection using ISS suggests that false positives are limited and we interpret the overlap of glutamatergic and GABAergic markers to reflect noise in the co-localization estimation derived from the misassignment of transcripts located in cell processes overlapping local cell bodies, although the presence of neurons that co-release glutamate and GABA has not been sufficiently systematically examined to be categorically ruled out. Co-localization of excitatory and inhibitory markers with *Nos1* and *Tac2*, two genes that mark columns within the dPAG, revealed that 57% of *Nos1+* cells were excitatory (93/164 cells), while 31% and 63% of *Tac2+* cells were excitatory and inhibitory, respectively (**Fig 1C**, **S3 Fig**). These data demonstrate that there are diverse subsets of spatially segregated excitatory neurons in dPAG and suggest that one or more of them may be responsible for the reported ability of optogenetic stimulation of excitatory neurons in dPAG to elicit flight.

### Optogenetic activation of excitatory neurons elicits goal-direct flight

Next, we performed optogenetic activation of selected neuron types in dPAG in order to determine whether they were capable of promoting flight behavior. Previous studies had reported that ChR2 activation of *Vglut2+*, but not *Gad2+* neurons in dPAG elicited a combination of

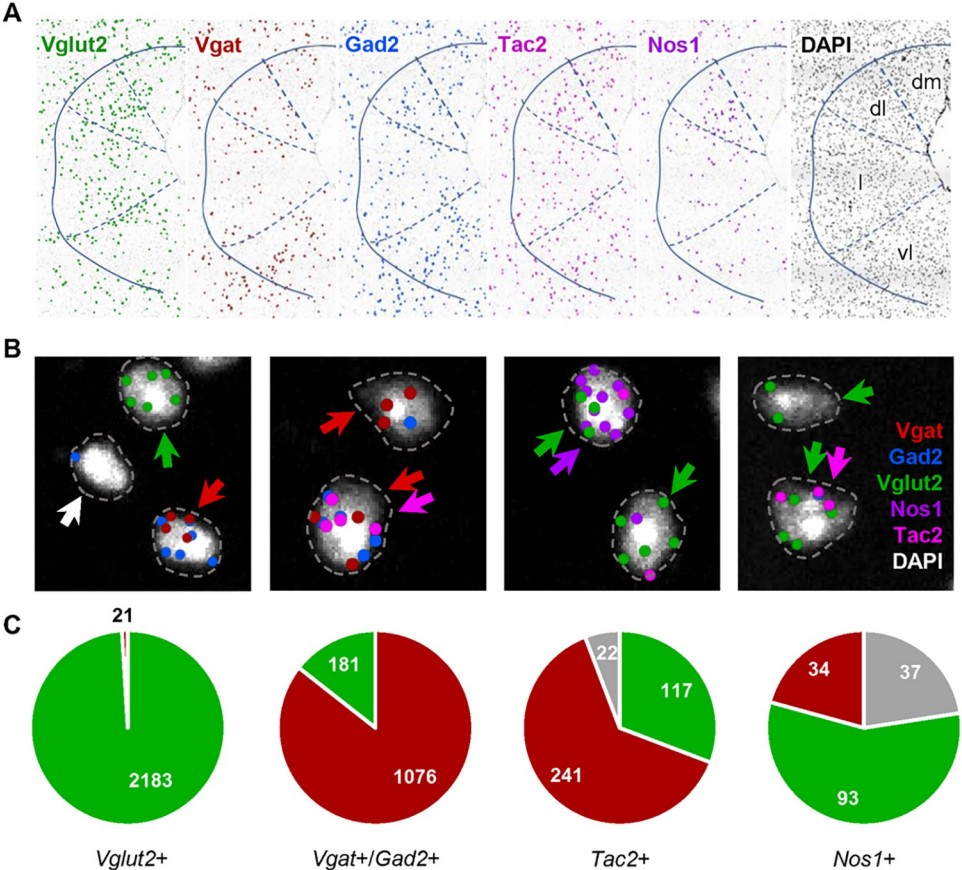

**Fig 1. Colocalization of *Nos1* and *Tac2* with excitatory and inhibitory cells in PAG. (A)** Localization of *Vglut2*, *Vgat*, *Gad2*, *Nos1*, and *Tac2* transcripts in PAG using multiplex *in situ* sequencing (ISS) showed regional patterns consistent with prior single gene *in situ* hybridization methods. DAPI signal was used to identify isolated cells and assign amplified ISS signals to putative single cells. The location of isolated cells that satisfied quality control criteria are represented by a single dot. **(B)** Representative images of cells classified as glutamatergic (red arrow), GABAergic (green arrow), *Tac2*+ (pink arrow), *Nos1*+ (purple arrow), or unclassified (white arrow). Genes were considered present if they showed at least two high quality (score > 2) spots (*Vgat* and *Gad2* were considered equivalent) that fell within an expanded perimeter of DAPI signal (dotted line). **(C)** Distribution of co-expression of glutamatergic and GABAergic markers and their co-localization with *Nos1* and *Tac2* across three replicates in dPAG.

freezing and flight, with the later behavior dominating at higher stimulation intensity [4, 22, 25]. To confirm and extend these findings, mice carrying either *Vglut2*::Cre, *Gad2*::Cre, *Tac2*::Cre, *Adcyap*::Cre, or *Nos1*::Cre were bilaterally injected with adeno-associated-virus expressing Cre-dependent ChR2 (AAV-*hSyn*::DIO-ChR2-YFP) or control virus (AAV-*hSyn*::YFP) into dPAG followed by the surgical placement of optical fibers over the injection areas (**Fig 2A**; **S1A and S1C Fig**). Light stimulation of mice was performed in a novel chamber containing a cardboard shelter following 5 minutes of habituation. Light stimulation of ChR2-expressing animals carrying the *Vglut2*::Cre and *Adcyap*::Cre transgenes showed clear freezing and flight responses when compared to YFP-expressing control animals, while those of mice carrying *Gad2*::Cre, *Tac2*::Cre, *Nos1*::Cre transgenes did not show any detectable behavioral response. Flight responses in *Vglut2*::Cre mice were dose-dependent, as they showed greater intensity with increased light power or frequency (**Fig 2B and 2C**). Latency to initiate flight typically occurred within one second following stimulation onset, but, unlike flight intensity, was not significantly affected by light frequency (**Fig 2D and 2E**). Notably, light-evoked flight in *Vglut2*::Cre infected mice resulted in a rapid retreat into the shelter, with mice

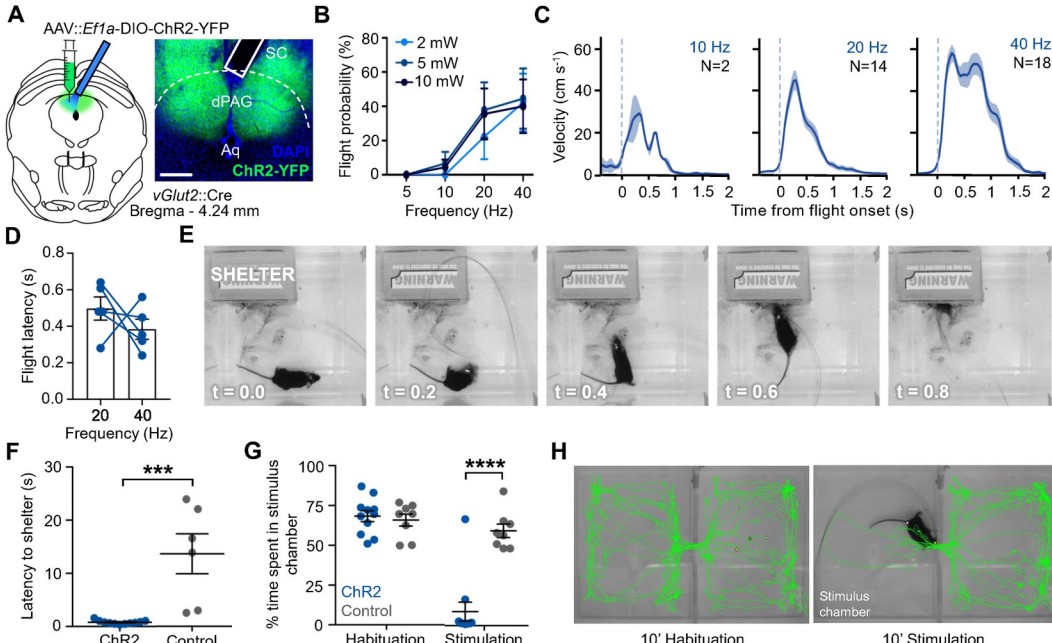

**Fig 2. Optogenetic stimulation of dPAG excitatory neurons elicits escape. (A)** Graphical representation of experimental strategy for optogenetic activation of excitatory cells in dPAG and representative histology of virus expression and fibre placement (solid line; SC, superior colliculus; dPAG, dorsal periaqueductal grey; Aq, aqueduct; scale bar, 250 μm). **(B)** Probability of observing flight upon optical stimulation of *Vglut2+* neurons in dPAG with increasing frequency and intensity (ChR2-expressing mice, N = 9, 5 trials per animal). **(C)** Evoked velocity aligned to flight onset (t = 0) upon optical stimulation at 10 mW at three different frequencies (N = number of trials). **(D)** Latency to elicit flight following optical stimulation (10 mW). Comparison between low and high frequency showed no significant difference (each blue dot represents a single subject). **(E)** Representative example of escape to shelter upon light stimulation in a ChR2-expressing animal (t = 0 indicates beginning of stimulation). **(F)** ChR2-expressing animals showed a short latency to escape to the shelter upon optical stimulation compared to the control group (ChR2 animals: N = 11, YFP animals: N = 6; Mann-Whitney unpaired t test, P = 0.0002). **(G)** ChR2-expressing animals avoided significantly more the stimulation chamber during the stimulation epoch when compared to YFP-expressing controls (ChR2: N = 11, YFP: N = 8; multiple t-test with Holm Sidak *post hoc*, t = 6.44, adjusted P < 0.0001). **(H)** Path of a representative ChR2-expressing animal during the habituation (left) and stimulation (right) epoch in the real-time place preference test; stimulation chamber is on the left.

turning and running toward the shelter where they then remained for an extended period (Fig 2E). Light stimulation of ChR2-expressing mice led to a significantly shorter latency to return to the shelter when compared to YFP-expressing control animals (Fig 2F).

To determine whether stimulation of dPAG neurons that elicit flight was aversive, mice carrying the *Vglut2*::Cre transgene were infected bilaterally with ChR2 or YFP control expressing viruses and surgically implanted with optic fibers above the infection site as above and allowed to habituate to a dual chamber real-time place preference (RTPP) apparatus. Following habituation, light was delivered (20 Hz, 10 mW) to the brain while the animal was in the preferred side of the apparatus. In nearly all animals expressing ChR2 light stimulation elicited escape to the opposite chamber and a failure to return to the stimulus chamber that resulted in a significant reduction of total time spent in the stimulus chamber when compared to YFP-expressing control animals (Fig 2G and 2H). These data indicate that optogenetic activation of excitatory neurons in dPAG is aversive.

## Projections to the cuneiform nucleus elicit goal-directed flight

The dorsal PAG projects to a set of downstream brainstem structures implicated in locomotion control, including lateral paragigantocellularis nucleus (LPGi), cuneiform nucleus (CnF),

superior colliculus (SC), lateral parabrachial nucleus (LPBS) and pedunculopontine nucleus (PPN) [28–30]. In particular, stimulation of CnF has been shown to evoke high-speed running [30–32] while LPGi was found to harbor excitatory and inhibitory neuron populations that promote and inhibit locomotion, respectively [36]. To test whether dPAG excitatory neurons project to CnF we delivered the presynaptic fluorescent fusion protein Synaptophysin-iRFP to *Vglut2*+ neurons in dPAG. An examination of iRFP signal in these animals revealed prominent presynaptic boutons in both CnF and the immediately ventral LPBS, but not in the core LPB or surrounding inferior colliculus (IC; **Fig 3A**). To determine whether projections from dPAG to CnF derived from a particular column or cell-type, we combined deposition of the retrograde fluorescent tracer cholera toxin B (CTB) in CnF and AAV-mediated fluorescent tagging of excitatory and inhibitory neurons in dPAG (**Fig 3B**). Fluorescent retrograde signal was restricted to the dorsolateral PAG consistent with earlier data using non-fluorescent anterograde tracers [37, 38] (**Fig 3C**). Colocalization of retrograde fluorescent signal with cell-type restricted expression of the mCherry fluorophore revealed the majority of projection neurons to be excitatory (*Vglut2*+: 74.9%, N = 3) and a minority inhibitory (*Gad2*+: 21.1%, N = 3). These data demonstrate that both excitatory and inhibitory neurons in the dorsolateral PAG project to CnF (**Fig 3D**).

Finally, we examined whether the projection from PAG to CnF could elicit goal-directed flight behavior. Cre-dependent ChR2 or YFP-expressing viruses were unilaterally delivered to dPAG, while a retrograde transported Cre-expressing virus was co-delivered unilaterally into CnF together with an iRFP-expressing virus to delineate the infection zone (**Fig 4A**). Because AAV-mediated infection of axons is less narrowly localized to the injection site than CTB, retrograde labeling in dPAG was more widespread (**Fig 3C** vs. **Fig 4B**). Optogenetic activation of retrograde ChR2-expressing neurons in dPAG elicited flight behavior in a dose-dependent manner (**Fig 4C**) and with latencies of less than one second (**Fig 4D**). Notably, flight responses elicited by light stimulation of ChR2-expressing dPAG projection neurons were directed toward the shelter (**Fig 4E**) and showed a significantly lower latency to escape than YFP-expressing control animals (**Fig 4F**).

## Discussion

Earlier cell-type specific optogenetic stimulation studies established that excitatory, but not inhibitory neurons in dPAG elicited flight behavior and here we have confirmed these findings and extended them to demonstrate that this behavior is directed toward a familiar shelter and can be elicited by a subset of dPAG neurons that project to CnF. Furthermore, we have used multiplex *in situ* sequencing to colocalize a set of selected marker genes in dPAG to define subsets of excitatory neurons (*Nos1*+ vs. *Tac2*+ vs. non-*Nos1*+/*Tac2*+) and used cell-type specific optogenetic activation to test their functional capacity to drive flight behavior. Our findings show that two major excitatory neuronal cell-types in dPAG–marked by *Nos1* and *Tac2* co-expression–are not able to drive flight behavior, at least when activated alone, suggesting that the remaining non-*Nos1*+/*Tac2*+ subpopulation is the key driver of flight. The finding that optogenetic activation of *Nos1*+ cells in dPAG is not sufficient to elicit defensive behavior is somewhat surprising given that pharmacological activation or inhibition of nitric-oxide synthase increases and decreases defensive responses to predator [39, 40]. However, the lack of spontaneous flight behavior upon optogenetic activation does not rule out that these cells are nevertheless involved in the modulation of defensive responses elicited by a natural threat and we interpret our findings to suggest only that *Nos1*+ cells are unlikely to include dPAG projection cells that drive the activation of downstream locomotor initiation centers. We note that, although *Tac2*+ cells in dPAG were not implicated in defensive behaviors here, stimulation of

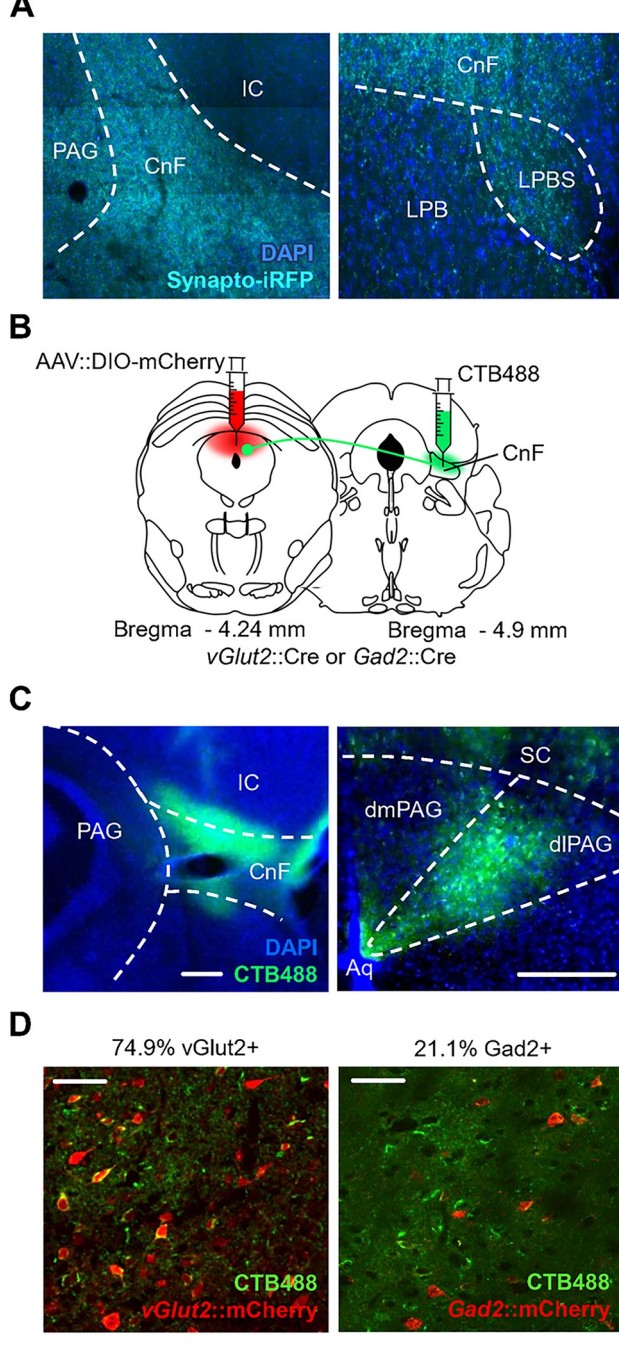

**Fig 3. Projections from dlPAG to the cuneiform nucleus. (A)** *Vglut2*+ cells in dPAG project to the cuneiform nucleus (CnF) and to the superior lateral parabrachial nucleus (LPBS). Cre-dependent Synaptophysin-iRFP expressing virus was injected in dPAG of a *Vglut2*::Cre transgenic animal. Sections show synaptic boutons expressing Synaptophysin-iRFP in CnF and LPBS (IC, Inferior Colliculus; Blue, DAPI). **(B)** Graphical representation of experimental strategy for retrograde labeling with CTB and identification of glutamatergic (*Vglut2*+) or GABAergic (*Gad2*+) identity of projection neurons. **(C)** Cell bodies of CnF afferents in PAG were limited to the dorsolateral column (green, CTB; left, CnF; right, PAG; blue, DAPI; scale bar, 250 μm). **(D)** CTB label (green) in PAG co-localized with both (left) glutamatergic (red) and (right) GABAergic (red) neurons (scale bar, 50 μm).

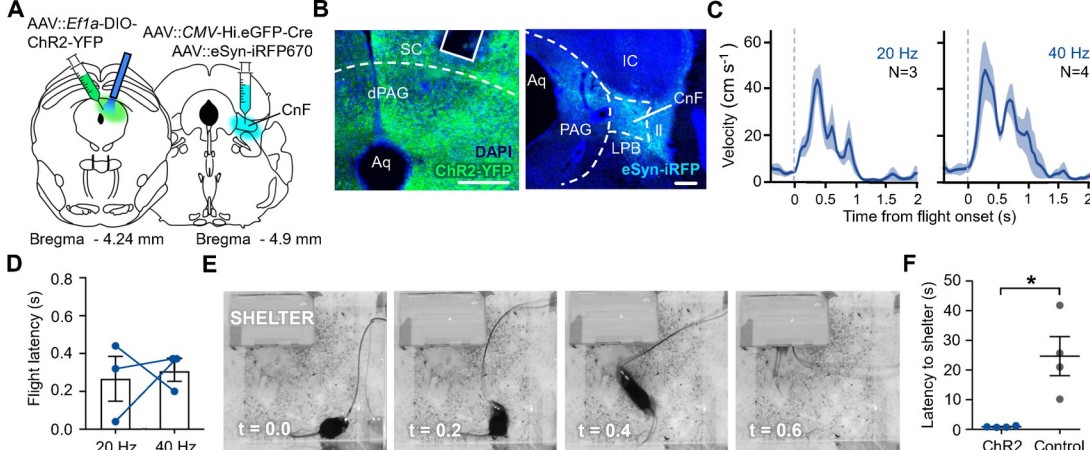

**Fig 4. Optogenetic stimulation of dPAG neurons that project to CnF elicits goal-directed flight. (A)** Graphical representation of experimental strategy for optogenetic activation of dPAG neurons that project to CnF. **(B)** Representative histology of Cre-dependent ChR2 expression (green) and fibre placement (solid line) in (left) dPAG and (right) site of retrograde viral injection in CnF (blue; SC, superior colliculus; CnF, cuneiform nucleus; LPB, lateral parabrachial nucleus; IC, inferior colliculus; ll, lateral lemniscus; scale bar, 200 μm). **(C)** Measured velocity aligned to flight onset at two different frequencies (N = number of trials, 1 trial per animal). **(D)** Latency to initiate flight from stimulation onset showed no significant difference between low and high frequency (blue dots represent individual subjects). **(E)** Representative example of escape to the shelter upon stimulation in a ChR2-expressing animal (t = 0 indicates stimulation onset). **(F)** ChR2-expressing animals showed significantly lower latency to escape in the shelter upon stimulation compared to control animals (N = 4; two-tailed unpaired t test, t (6) = -3.63, *P = 0.0110).

*Tac2*+ cells in forebrain regions have been shown to elicit avoidance [41]. Our findings leave open the question of the role of *Tac2*+ cells and the dmPAG more generally in defensive behavior. We note for example that dmPAG is a direct target of frontal cortical inputs that are known to inhibit defensive responses to threat [5, 42]. Again, our optogenetic stimulation approach was not able to determine whether *Tac2*+ cells might be involved in the modulation of flight–for example, via its sensory-dependent triggering, disinhibition, or termination. We also note that our study focused exclusively on the capacity to elicit flight behavior and thus we cannot draw conclusions about other defense-related outputs linked to PAG such as analgesia.

Flight-like behavior can be elicited by electrical or optogenetic stimulation of a series of sub-cortical brain structures that can be assigned to two pathways as defined by functional neuro-anatomy and tract tracing experiments. The first pathway passes via the medial hypothalamic defensive system (AHN, VMHdm, PMD), its major targets in dPAG and on to MLR (consisting of PPN and CnF) and from there to excitatory neurons in LPGi that project to the ventral spinal cord to produce high speed locomotion [3, 24, 36, 43]. The second pathway starts from visual and auditory responsive areas in superficial SC and IC and passes to deep layers of SC and from there to both dPAG and MLR where it joins the first pathway [44, 45]. A major finding of our work is that projections from dPAG to CnF elicit goal-directed flight behavior. Optogenetic stimulation of CnF neurons is known to elicit high-speed locomotion and projections from dPAG to CnF had been suspected to mediate dPAG-dependent flight behavior [46]. Our findings confirm this hypothesis. Critically, the observation that animals in which CnF-projecting neurons in dPAG were optogenetically stimulated showed robust goal-directed flight suggests that critical spatial information required for remembering, orienting to, and actively tracking shelter position reaches the defensive escape pathway downstream of dPAG, possibly as a result of converging afferents from sensory-receptive areas in superior and inferior colliculus and cortex in the MLR. However, our findings do not allow us to rule

out that descending or ascending collaterals of these dPAG neuron types might also have a role in supporting flight. Future work on the functional integration of dPAG projections to CnF and downstream locomotor areas such as LPGi [36] will be needed to understand how high-speed escape is modulated to guide the animal to safety and how this is coordinated with cortical inputs contributing associative models of the environment.

## Materials & methods

### Mice

All experimental subjects were adult male C57BL/6 mice obtained from local EMBL or EMMA colonies or from Charles River Laboratories (Calco, Italy). For cell-type specific opto-genetic manipulation *Vglut2*::Cre [47], *PACAP*::Cre [48], *Tac2*::Cre, *Gad2*::Cre [49] and *Nos1*::CreERT2 [49] were used. Where necessary mice were treated with tamoxifen for five consecutive days during the second week of recovery from surgery (Sigma, 40 mg/kg i.p.). All mice were genotyped before experimentation. For retrograde projection activation wild-type mice were used. Mice were maintained in a temperature (22±1˚C) and humidity-controlled (50% rH) facility on a 12 h light-dark cycle (lights on at 07:00) with food and water provided *ad libitum*.

### Animal surgery

Isoflurane (induction 3%, maintenance 1.5%; Provet) in oxygen-enriched air was used to anaesthetize mice fixed in a stereotactic frame (Kopf Instruments). All mice received a subcutaneous injection of 1% Caprofen solution (5 mg/kg Rymadil, 0.01 ml/g) for surgical analgesia. For cell-type specific activation, *Vglut2*::Cre, *Adcyap1*::Cre, *Tac2*::Cre and *Nos1*::Cre, *Gad2*::Cre transgenic mice were infused bilaterally in dPAG (AP:-4.24, L:±0.30, DV:-2.15 from Bregma) with 60–120 nl/side of AAV5-*Ef1a*::DIO-hChR2(E123T/T159C)-EYFP virus (UNC Vector Core) for experimental groups or AAV5-*Ef1a*::DIO-EYFP virus (UNC Vector Core) for control groups using a pulled glass capillary. In the same surgery mice were unilaterally implanted with custom-built fibre connectors (fibre: 0.66 numerical aperture, 200 μm core diameter; ceramic ferrule: 230 μm internal diameter, 1.25 mm outer diameter; Prizmatix) in the dPAG just above the viral infection site (AP:-4.24, L:+0.90, DV:-2.15 from Bregma, at 26˚ angle to avoid sigmoid sinus damage). For retrograde afferent activation wild-type mice were injected unilaterally in CnF (AP:-5.7 L:±1.23, DV:-3.80 from Bregma, with a posterior-anterior 15˚ angle) with 180 nl of virus solution composed of AAV-*CMV*::Hi.eGFP-Cre (Penn Vector Core) mixed with AAV-*eSyn*::iRFP670 as a marker for the site of injection (UNC Vector Core) at a ratio of 4:1. In the same surgery animals were unilaterally injected in the dPAG with 180 nl of AAV5-*Ef1a*::DIO-hChR2(E123T/T159C)-EYFP and AAV5-*hSyn*::DIO-mCherry as a marker for the site of injection (UNC Vector Core) at 4:1 ratio. Mice in the control group were injected, using the same coordinates, with AAV5-*Ef1a*::DIO-EYFP and AAV5-*hSyn*::DIO-mCherry in a ratio of 4:1. The dPAG injection was performed by insertion of the capillary in the contralateral hemisphere of the target area with a 28˚ mediolateral angle (AP: -4.24, ML: +0.85, DV: -2.30 from Bregma) to avoid infection of the ipsilateral superior colliculus. The optic fibre (0.22 numerical aperture, 225 μm core diameter; ceramic ferrule: 1.25 mm outer diameter; Thorlabs) was implanted unilaterally over the ipsilateral dPAG (AP: -4.24, ML: -1.45, DV: -2.00 from Bregma with a 26˚ mediolateral angle). In both the cell-type specific dPAG activation experiment and the retrograde projection activation experiment injections were at a rate of approximately 60 nl/min. At the end of surgical procedures animals were injected with 1 ml saline (i.p.) and remained on a controlled temperature heating pad overnight. Animals were given paracetamol (0.8 mg/ml, Tachipirina) in the drinking water for 3

days and monitored for one week to assess normal recovery. After surgical procedure mice were singly housed to avoid implant damage due to conspecific interactions.

## Optogenetic stimulation

In the cell-type specific dPAG activation experiments optical stimulation of ChR2-expressing cells was carried out by delivery of blue light (465 nm) from LED modules attached to a rotary joint via high performance patch cables (Plexbright, Plexon). All stimulation trains were generated with Radiant V2.2 (Plexon). For optical stimulation of ChR2-expressing cells in the retrograde projection activation experiment, blue (465 nm) laser light (PSU-III-LED, Thorlabs) was applied. Light was delivered from the laser via high performance patch cables (Thorlabs); all stimulation trains were generated with Pulser Software. Stimulation pulse width duration was fixed at 15 ms. Power intensities and frequencies for the experimental group were picked for each subject as the minimum value evoking the optimal behavioral response (sudden burst of locomotor activity, 10–20 mW). Control animals were stimulated at the highest power and frequency (10 mW, 20 Hz for cell-type specific dPAG activation; 20 mW, 40 Hz for retrograde projection activation).

## Behavior

All behavioral experiments were performed on mice at least 8 weeks old. Behavioral tests were performed 3–4 weeks after surgery. All behavioral testing occurred during animals' light cycle. All behavioral videos were recorded with a top view camera and Biobserve Viewer under ambient lighting. Animals were handled for at least 2 days before the start of any behavioral assay. All mice were handled according to protocols approved by the Italian Ministry of Health (#137/2011-B, #231/2011-B and #541/2015-PR) and commensurate with NIH guidelines for the ethical treatment of animals.

**Locomotion assay.** To assay overt locomotor behavior in response to optogenetic stimulation mice were attached to optical patch cables and placed in a novel transparent plexiglass chamber (24 x 24 x 24 cm) and allowed to habituate for 5 min. After the habituation phase, *Vglut2*::Cre and *PACAP*::Cre mice infected in dPAG with ChR2-expressing AAV (AAV-*Ef1a*::DIO-CHR2-YFP) and implanted with optic fibers above the infection site, received 1 s long light stimulation followed by a 60 s inter-stimulation interval. Stimulation intensity was selected according to an increasing pattern (2/5/10 mW) each applied at 5/10/20/40 Hz in an increasing sequence. Each stimulation combination was repeated five times. Because no overt locomotor response was detected in *Tac2*::Cre, *Nos1*::Cre, and *Gad2*::Cre mice stimulation lasted 120 s at each intensity and frequency, followed by 120 s inter-stimulus interval in order to assess any potential subtle responses or long-term post-stimulation effects. Each stimulation condition was repeated five times. For retrograde projection activation stimulation lasted 1 s at each intensity and frequency, followed by 60 s inter-stimulus interval. Stimulation intensity followed an increasing pattern (0.5–25 mW) applied at 20 Hz and then, from 10 mW, repeated at 40 Hz.

**Goal-directed escape assay.** Animals were placed in a transparent plexiglass chamber (24 x 24 x 24 cm) with a shelter (18 x 7 x 13 cm cardboard box with large opening on one side) placed in a corner. The shelter floor was lined with the animal's home cage bedding. Animals were free to explore and habituate for 5 min or until they displayed no reluctance to enter the shelter. After habituation, light was delivered at varying intensity and frequency (10 mW, 20 Hz for dPAG *Vglut2*::Cre activation; 10–20 mW, 40 Hz for retrograde projection activation) when the animal was outside the shelter and at the far side of the cage and persisted until the animal entered the shelter. The procedure was repeated 3–6 times.

**Real time place preference.** Animals were placed in an apparatus composed of two chambers (each 24 x 24 x 24 cm) connected by a door (3 cm width) and were free to access both chambers for ten minutes. At the end of the habituation phase the preferred chamber was selected as the stimulation chamber and for the following 10 minutes the animal was stimulated (10 mW, 20 Hz) whenever it entered that chamber. If the animal remained in the chamber for more than 90 s the stimulation was terminated for 30 s before initiating another 90 s stimulation bout. Stimulation was terminated upon exit from the chamber.

## Afferent mapping with CTB

For identification of projection neurons in PAG, *Vglut2*::Cre or *Gad2*::Cre mice were injected bilaterally in dPAG with AAV-*hSyn*::DIO-mCherry (150 nl each side) and after one week injected with 60 nl 0.5% CTB-488 (Thermo Fisher #C34775) unilaterally into the CnF (AP: -4.90, L: 1.23, DV: -3.00 from Bregma) and the capillary left in place for at least 10 min to avoid CTB spread along the injection tract. Animals were perfused one week after CTB injection.

## *In situ* sequencing

CARTANA (10x Genomics) kits were used to process samples for *in situ* sequencing following manufacturer instructions. CARTANA were provided a list of genes and designed and provided the padlock probes. Imaging of fluorescent samples was carried out with an X-light V3 spinning disk (Crest Optics) coupled to a Nikon Ti inverted microscope (Nikon), a Prime BSI sCMOS camera (Photometrics), and a Celesta laser light engine (Lumencor). Images were acquired in 5 channels using excitations at 405, 477, 546, 638 and 749 nm. Appropriate filters for DAPI, GFP, Cy3, Cy5 and Cy7 were used. Imaging was done using a 20x NA 0.8 Nikon objective. Z-stacks over the region of interest were acquired using a custom JOB program created in Nikon NIS Elements. Z-stacks were converted to a single image by maximum intensity projection within NIS Elements. Image registration was done in two steps, using custom tools in MATLAB (The Mathworks) [50]. First, a coarse registration was performed by fast Fourier transform using a downscaled image of the DAPI channel. Then, full resolution images were registered using the intensity and the image of the dots themselves in squared tiles of 500 pixels and re-stitched together. Decoding was performed using the ISTDECO algorithm that combines spectral and spatial information to decode the identity and position of transcripts. Two fake codes were included in the codebook to control for possible artifacts. Fake codes comprised less than 2% of all decoded spots. We used squared tiles of 500 pixels, a PSF size of 2, an intensity percentile value of 99.95 and a quality threshold of 0.5. The DAPI signal was used to segment nuclei using StarDist [51] and the cell boundaries were expanded using inbuilt function in QuPath. Gene expression signals (spots) were thresholded (quality score > 1.3) and only those spots falling within expanded, but well segmented cells (manually selected; dotted lines in **Fig 1B**) were retained for co-expression analysis. An arbitrary cutoff of two spots was used to call gene expression for estimating cell-types; Gad2 and Vgat signal were collapsed before scoring spots for GABAergic identity. Map2 and Grin2b were not used in co-expression analysis. A mask based on macroscopic DAPI staining and brain atlas boundaries was used to restrict the analysis to dPAG. Both hemispheres of the brain from three distinct sections (along the rostro-caudal axis) were used for co-expression analysis, although only one half is shown in (**Fig 1A**, **S2 Fig**).

## Histology and microscopy

Mice were transcardially perfused with PBS followed by 4% paraformaldehyde in PB. Brains were left to postfix overnight in 4% PFA at 4˚C. Brains were either sectioned in PBS with a

vibratome (Leica VT1000s) or cryo-sectioned. For cryo-sectioning brains were first briefly rinsed in PBS after post-fixation, then left in 30% sucrose in PBS for 2 days before flash freezing in pre-chilled isopentane. Frozen brains were sectioned on a cryostat (Leica CM3050s). Sections of 50 or 70 μm were taken from the areas of interest. DAPI (5 mg/ml) was added directly to the mounting medium (MOWIOL). Widefield images were acquired with a Leica LMD5 microscope; confocal images were acquired with a Leica SP5 with resonant scanning. Optic fibre placements and injection sites were verified according to anatomical landmarks and an atlas (Franklin and Paxinos, 2008) and widefield images (Leica LMD5). Mice with incorrect fibre position or unsuccessful viral infection were excluded from the analysis.

### Data analysis

Behavioral data were obtained with either manual scoring software (Solomon Coder) or automatic scoring software (Biobserver Viewer and Bonsai). For manual scoring flight was defined as a "sudden burst in velocity".

### Statistics

All statistical analysis was performed using Prism 7 (GraphPad). All p-values were adjusted and error bars were mean±SEM unless otherwise noted. Group differences were determined using multiple t-test with Holm-Sidak *post hoc* tests, Mann-Whitney unpaired t-test, or two-way ANOVA with Tukey's *post hoc* testing.

### Supporting information

**S1 Fig.** (**A-C**) Graphical representation of experimental strategy for optogenetic activation of cell-types in dPAG. (**B-D**) Representative histology showing ChR2 expression (green) and fibre placement for each Cre driver line (solid line; scale bar, 250 μm).
(TIF)

**S2 Fig.** (**A-B**) Distribution of excitatory, inhibitory, *Nos1* and *Tac2* cells in PAG. Localization of *Vglut2*, *Vgat*, *Gad2*, *Nos1*, and *Tac2* transcripts in PAG using multiplex *in situ* sequencing in two independent brain sections.
(TIF)

**S3 Fig.** (**A-C**) Distribution of co-expression of glutamatergic and GABAergic markers and their co-localization with *Nos1* and *Tac2* in three independent brain sections.
(TIF)

### Acknowledgments

We thank Francesca Zonfrillo, Claudia Valeri, Roberto Voci, and Valerio Rossi for support with animal husbandry and management, Quifu Ma for sharing *Tac2*::Cre mice, and Angelo Raggioli for support with production of viruses and the EMBL Microscopy, Histology, Gene Editing & Embryology, and Laboratory Animal Facilities for support.

### Author Contributions

**Conceptualization:** Emmy Tsang, Cornelius T. Gross.

**Formal analysis:** Emmy Tsang, Camilla Orlandini, Rahul Sureka, Alvaro H. Crevenna, Izzie Prankerd, Maria E. Masferrer.

**Funding acquisition:** Cornelius T. Gross.

**Investigation:** Emmy Tsang, Camilla Orlandini, Alvaro H. Crevenna, Emerald Perlas, Izzie Prankerd, Maria E. Masferrer.

**Methodology:** Emmy Tsang, Camilla Orlandini, Rahul Sureka, Alvaro H. Crevenna, Emerald Perlas, Izzie Prankerd, Maria E. Masferrer.

**Project administration:** Cornelius T. Gross.

**Resources:** Cornelius T. Gross.

**Supervision:** Cornelius T. Gross.

**Validation:** Rahul Sureka.

**Writing – original draft:** Camilla Orlandini, Rahul Sureka, Maria E. Masferrer, Cornelius T. Gross.

**Writing – review & editing:** Rahul Sureka, Cornelius T. Gross.

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
