## [Decision Letter · Decision Letter 0]

30 Aug 2022

PONE-D-22-23069Induction of flight via midbrain projections to the cuneiform nucleusPLOS ONE

Dear Dr. Gross,

Thank you for submitting your manuscript to PLOS ONE. After careful consideration, we feel that it has merit but does not fully meet PLOS ONE’s publication criteria as it currently stands. Therefore, we invite you to submit a revised version of the manuscript that addresses the points raised during the review process.

There are several positive features of the present study which include the methodologies employed: optogenetic stimulation to elicit flight behavior and *in situ* hybridization to characterize the neurochemical properties of the neurons in the region where stimulation was employed. As a result, the authors provide new and potentially important information to the existing literature in this field of investigation. However, the reviewer suggests that there are a number of issues that should be addressed in order to strengthen the manuscript. These are indicated below.

1.The figures provided make it difficult to know exactly which areas are activated following stimulation. Further analysis might allow for a better understanding of this matter.

2.There is no attempt to map the region of the PAG where flight can be elicited and where it is not present. If the authors have data along these lines or can obtain such information, that would be helpful.

3.There was no discussion concerning other brain regions such as hypothalamus where similar responses have been reported upon stimulation. And such information should he added to the Discussion section in an attempt to synthesize our understanding of the neurobiology of this behavior.

4.When stimulation is applied to the PAG eliciting flight, what pathways are activated and possibly inhibited (e.g., descending pain inhibitory pathway to spinal cord). Said otherwise, it would be helpful if the authors were to include the putative pathways mediating this pathway. Is the PAG for this behavior the final neuronal region in the descending pathway for elicitation of flight or do others exist? Is there an ascending component? Moreover, the regions mediating flight behavior seem to overlap with those mediating defensive behavior. Any thoughts on why stimulation did not elicit defensive behavior at least on some occasions? The maps shown from studies in cat and rat indicate that the dorsolateral PAG and medial hypothalamus are central sites where defensive behavior are elicited and clearly overlap with those regions of the PAG described in the present manuscript.  Or perhaps, is there a species specificity explanation here?

5.The Discussion regarding how flight behavior may be linked to `’locomotor`’ responses is not clear and needs to be clarified.

6.Perhaps the most questionable aspect of the study related to the conclusions suggesting a direct projection from the PAG to the cuneiform nucleus, The problem here is that evidence for the use of this method as an anatomical tracing method seems to be lacking in the literature. Moreover, while such a projection may very well exist, how does one know that activation of such a pathway is related to the expression of flight behavior. Here, one way of possibly addressing this issue would be to test whether stimulation of the cuneiform nucleus would also elicit this behavior. Alternatively, do the authors have any data showing that stimulation of sites adjoining those which elicit flight do not show activation of the cuneiform nucleus. Such data would also serve as an effective control for this aspect of the study.

We look forward to receiving your revised manuscript.

Kind regards,

Allan Siegel

Academic Editor

PLOS ONE

Journal Requirements:

"The work was supported by EMBL (https://www.embl.org/) and the European Research Council (ERC) Advanced Grant (https://erc.europa.eu/funding/advanced-grants) COREFEAR to C.T.G."

"The work was supported by EMBL (https://www.embl.org/) and the European Research Council (ERC) Advanced Grant (https://erc.europa.eu/funding/advanced-grants) COREFEAR to C.T.G." 

Reviewers' comments:

Reviewer's Responses to Questions

**Comments to the Author**

1. Is the manuscript technically sound, and do the data support the conclusions?

Reviewer #1: Yes

2. Has the statistical analysis been performed appropriately and rigorously? 

Reviewer #1: Yes

3. Have the authors made all data underlying the findings in their manuscript fully available?

Reviewer #1: Yes

4. Is the manuscript presented in an intelligible fashion and written in standard English?

Reviewer #1: Yes

5. Review Comments to the Author

Reviewer #1: The present study utilized optogenetic stimulation of the periaqueductal gray (PAG) of mice to firstly identify selected sites eliciting flight and to characterize the neurochemical properties of neurons in that region utilizing in situ hybridization. Secondly, the study further attempts to use these methodologies to identify a PAG efferent pathway to the cuneiform n.

The first aspect of the study is excellent. It utilizes optogenetic activation to induce flight responses where the responses are induced by stimulation of cell bodies and not fibers of passage, which were always a matter of concern with studies conducted in earlier periods. The labelling of the neurochemical properties of the neurons in the vicinity where stimulation was applied reflects another positive feature of this study. Collectively, they provide new and interesting information to the study of the neurobiology of flight behavior.

Nevertheless, with respect to this phase of the study, the manuscript could have addressed a number of other aspects which would have strengthened the paper. For one, it is difficult to discern from the figures the precise areas activated from stimulation at a given time. Second, there was no evidence of mapping of the region of the PAG associate with this behavior. For example, how far along the rostral-caudal length of the PAG could the response be elicited? Likewise, was this behavior limited to the dorsolateral aspect of the PAG and not to the ventral quadrant? Thirdly, there was no mention in the manuscript of other regions of the brain where flight can be elicited (e.g., hypothalamus), in cat and rat and possibly other species. Fourth, there is no mention in the Discussion of the descending pain inhibitory pathway from the PAG to spinal cord and whether its actions are blocked by stimulation of regions eliciting flight. Fifth, there is no clear discussion of the proposed output pathways to the lower brainstem and beyond mediating flight and possible ascending pathways to the forebrain where integration of this response might also take place. Moreover, is the dorsolateral PAG the final region of integration of the response or whether there are other sites involved such as pontine and medullary nuclei. It is not clear how flight behavior can be lumped together with “locomotor” responses as opposed to its being a unique response in itself. Do the authors have any thoughts on how the possible circuit mediating flight might be distinguished from that mediating defensive behavior since both behaviors are associated with closely related regions of the brainstem and hypothalamus.

Conclusions about neuroanatomical connections derived from this methodology are drawn from the second aspect of the study. Here, I would argue that the conclusions reached may possibly be going beyond what the observations and methodology would allow. From what was shown in the figures, it is hard to tell whether one is looking at an actual region activated (i.e., cuneiform nucleus) by virtue of a direct anatomical connection, or by diffuse activation within the region of stimulation, especially since the nucleus is situated proximal to the PAG. In this regard, conclusions regarding the observations reported in this manuscript could have been supplemented by more traditional neuroanatomical methods such as retrograde labelling. Further, how does one know that the so-called activation of the cuneiform nucleus is functionally related to flight. Was any attempt made to apply stimulation to this nucleus? If not, is it possible that activation of this nucleus was unrelated to the process in question?

In summary, the strength of the manuscript is in its first phase and that should be highlighted and in the absence of additional neuroanatomical experiments, conclusions regarding direct anatomical connections between the PAG and cuneiform nucleus with respect to flight in mice remain somewhat questionable at best and should be expressed with greater caution.

6. PLOS authors have the option to publish the peer review history of their article (what does this mean?). If published, this will include your full peer review and any attached files.

Reviewer #1: **Yes: **Allan Siegel

---

## [Author Response · Author response to Decision Letter 0]

4 Jan 2023

Point-by-point Rebuttal

Tsang et al. “Induction of flight via midbrain projections to the cuneiform nucleus”

Author note: changes to the main text have been highlighted in yellow for easy tracking. Rebuttals to each comment are indicated in italics below.

Reviewer and Editor Comments:

1.The figures provided make it difficult to know exactly which areas are activated following stimulation. Further analysis might allow for a better understanding of this matter.

In reviewer comments: For one, it is difficult to discern from the figures the precise areas activated from stimulation at a given time.

Authors’ response: Optic fibers for our optogenetic stimulation experiments were implanted to achieve a final location ~200 μm above dPAG at its mid-anterior/posterior point (at the level of the oculomotor nucleus) so as to achieve maximal activation of ChR2 in dPAG. Light intensity drops off as the distance from the fiber tip increases, and the half-maximal intensity depth was estimated to be 340 μm at 2 mW and 670 μm at 10 mW for the cell-type specific activation experiments and 1400 μm at 10 mW and 1800 μm at 20 mW for the retrograde cell body activation experiment (calculations based on: https://nicneuro.net/optogenetics-depth-calculator/). Given the relatively large size of the area receiving light in such experiments and the broad spread of AAV-delivered ChR2 expression typical in mouse brain (e.g. Figure 2A), we cannot localize the manipulation to neurons in any single PAG column, but rather can conclude that we achieved a stimulation of the relevant cell-type in dorsal PAG (dPAG) generally. Representative locations of the fiber tips as determined by post-experiment histology are shown for each experiment in the relevant figures (see Figure 2A & 4AB). 

2.There is no attempt to map the region of the PAG where flight can be elicited and where it is not present. If the authors have data along these lines or can obtain such information, that would be helpful.

In reviewer comments: Second, there was no evidence of mapping of the region of the PAG associate with this behavior. For example, how far along the rostral-caudal length of the PAG could the response be elicited? Likewise, was this behavior limited to the dorsolateral aspect of the PAG and not to the ventral quadrant?

Authors’ response: Our decision to study the dorsal PAG as an area whose stimulation elicits flight is based on extensive existing literature demonstrating the involvement of this part of PAG in active escape behaviors. Stimulation of ventrolateral PAG, on the other hand, elicits freezing and immobility, with optogenetic activation of vlPAG Vglut2+ neurons eliciting immobility and analgesia, for example (Tovote et al., 2016). Thus, our study focused on the role of dPAG cell-types and their projections on controlling flight and we did not attempt to confirm earlier studies showing that neighboring areas, such as vlPAG, do not elicit flight. As explained above, further dissection at the level of sub-dPAG columns (e.g. dmPAG, dlPAG or lPAG) would only have been possible with Cre-driver lines that are restricted to these areas. Indeed, Nos1::Cre and Tac2::Cre are enriched in dlPAG and dmPAG, respectively (Figure 1A), but given that these lines did not drive overt behavioral responses and these populations represent only a fraction of neurons in these columns, we cannot draw any definitive conclusions about the function of dPAG columns. As far as the relevant anterior/posterior position in dPAG, we did not explore this issue in depth as our aim was to focus on cell types and projections, rather than dPAG subregions in this study. 

3.There was no discussion concerning other brain regions such as hypothalamus where similar responses have been reported upon stimulation. And such information should he added to the Discussion section in an attempt to synthesize our understanding of the neurobiology of this behavior.

In reviewer comments: Thirdly, there was no mention in the manuscript of other regions of the brain where flight can be elicited (e.g., hypothalamus), in cat and rat and possibly other species. 

Authors’ response: Flight-like behavior can be elicited from a series of subcortical brain structures that can be assigned to two pathways as defined by functional neuroanatomy and tract tracing experiments. The first pathway responds to multisensory threat information that passes via the medial hypothalamic defensive system (AHN, VMHdm, PMD), its major targets in dPAG and on to MLR (consisting of PPN and CnF) and from there to excitatory neurons in LPGi that project to the ventral spinal cord to produce high speed locomotion (Canteras 2002; Gross & Canteras 2012; Capelli et al. 2017). The second pathway starts from visual and auditory areas in superficial SC and IC and passes to deep layers of SC and from there to both dPAG and MLR where it joins the first pathway (Branco & Redgrave 2020; Evans et al., 2018). Electrical or optogenetic stimulation of any of these nuclei across species can elicit flight. We have now added information on the regions shown to elicit flight in the Discussion section of our revised manuscript.

4.When stimulation is applied to the PAG eliciting flight, what pathways are activated and possibly inhibited (e.g., descending pain inhibitory pathway to spinal cord). Said otherwise, it would be helpful if the authors were to include the putative pathways mediating this pathway. Is the PAG for this behavior the final neuronal region in the descending pathway for elicitation of flight or do others exist? Is there an ascending component? Moreover, the regions mediating flight behavior seem to overlap with those mediating defensive behavior. Any thoughts on why stimulation did not elicit defensive behavior at least on some occasions? The maps shown from studies in cat and rat indicate that the dorsolateral PAG and medial hypothalamus are central sites where defensive behavior are elicited and clearly overlap with those regions of the PAG described in the present manuscript. Or perhaps, is there a species specificity explanation here?

In reviewer comments: Fourth, there is no mention in the Discussion of the descending pain inhibitory pathway from the PAG to spinal cord and whether its actions are blocked by stimulation of regions eliciting flight. Fifth, there is no clear discussion of the proposed output pathways to the lower brainstem and beyond mediating flight and possible ascending pathways to the forebrain where integration of this response might also take place. Moreover, is the dorsolateral PAG the final region of integration of the response or whether there are other sites involved such as pontine and medullary nuclei.

Do the authors have any thoughts on how the possible circuit mediating flight might be distinguished from that mediating defensive behavior since both behaviors are associated with closely related regions of the brainstem and hypothalamus.

Authors’ response: Stimulation of dPAG has been shown to elicit defensive behaviors, and our data corroborates these findings and extends them to specific cell-types in dPAG, including those projecting to CnF. In this study we did not examine the moderation of responses to painful stimuli and thus cannot comment on whether the cell populations whose activation elicits flight also moderate pain. However, previous studies have pointed to excitatory ventral PAG cells as the major PAG neuron class mediating nociceptive modulation (Tovote et al. 2016). We have now added a sentence in the Discussion to indicate this limitation.

The repertoire of an animal’s overt defensive behavior includes flight, freezing, defensive attack, and risk assessment (Blanchard et al. 1998; Blanchard and Blanchard 2008; Bolles 1970) – in that sense, flight is a key example of defensive behavior. These defensive behaviors are elicited and supported by the medial hypothalamic defensive system and its outputs in dPAG and downstream locomotor areas such as MLR, LPGi and ultimately motor pattern generators in spinal cord. These pathways are described in the introduction of the manuscript, which we requote here:

“How does activity in dPAG trigger high speed escape behavior? Anatomical tract tracing suggested that dPAG neurons project to several downstream brainstem structures known to produce locomotor behavior, including the mesencephalic locomotor region (MLR) and lateral paragigantocellularis nucleus (LPGi; Meller and Dennis, 1991). Recent cell-type specific optogenetic activation showed that a prominent target of dPAG in MLR, the cuneiform nucleus (CnF), is capable of producing high speed locomotor responses, raising the possibility that dPAG projections to CnF may provide a key output for defensive escape (Redgrave et al., 1988; Caggiano et al., 2018; Sandner et al., 1992; Dielenberg, Hunt, & McGregor, 2001).” 

In this study we have not examined the consequences of activating dPAG cells that have ascending projections, and we cannot comment on whether activating these is sufficient to elicit flight. A statement to this effect has been added to the Discussion. 

Finally, our study focused on understanding the role of the dPAG-CnF projection in flight behavior and we did not extend the study to determine the descending pathways mediating flight. However, the CnF has been shown to project directly to Vglut2+ medullary LPGi neurons and from there to the ventral spinal cord and this pathway is sufficient and necessary to elicit dose-dependent high speed locomotion (Capelli et al. 2017; see also Caggiano et al. 2018). We speculate that this pathway supports the flight responses we are able to elicit from dPAG cell classes. This reasoning is mentioned in the last paragraph of the Discussion section.

5.The Discussion regarding how flight behavior may be linked to `’locomotor`’ responses is not clear and needs to be clarified.

In reviewer’s comments: It is not clear how flight behavior can be lumped together with “locomotor” responses as opposed to its being a unique response in itself. 

Authors’ response: Flight is characterized by high-speed locomotion away from a threat and toward relative safety. In this work, we showed that stimulation of both dPAG Vglut2+ cells and CnF-projecting dPAG cells elicits high speed locomotion toward a shelter – a behavioral response that suggests that we are invoking defensive flight pathways in the brainstem and that is consistent with dPAG being required for flight responses to naturalistic threats such as predators. Earlier work has identified medullary (e.g. LPGi) and spinal cord circuits that support and drive high speed flight and showed that these neurons receive direct inputs from CnF and surrounding MLR structures (Caggiano et al. 2018). These anatomical connections lead us to speculate that the elicitation of flight-like high speed locomotion toward a shelter we observed depends on the activation of downstream medullary and spinal cord neurons that drive non-goal oriented high-speed locomotion. This reasoning is mentioned in the last paragraph of the Discussion section.

6.Perhaps the most questionable aspect of the study related to the conclusions suggesting a direct projection from the PAG to the cuneiform nucleus, The problem here is that evidence for the use of this method as an anatomical tracing method seems to be lacking in the literature. Moreover, while such a projection may very well exist, how does one know that activation of such a pathway is related to the expression of flight behavior. Here, one way of possibly addressing this issue would be to test whether stimulation of the cuneiform nucleus would also elicit this behavior. Alternatively, do the authors have any data showing that stimulation of sites adjoining those which elicit flight do not show activation of the cuneiform nucleus. Such data would also serve as an effective control for this aspect of the study.

In reviewer’s comments: From what was shown in the figures, it is hard to tell whether one is looking at an actual region activated (i.e., cuneiform nucleus) by virtue of a direct anatomical connection, or by diffuse activation within the region of stimulation, especially since the nucleus is situated proximal to the PAG. 

In this regard, conclusions regarding the observations reported in this manuscript could have been supplemented by more traditional neuroanatomical methods such as retrograde labeling. Further, how does one know that the so-called activation of the cuneiform nucleus is functionally related to flight. Was any attempt made to apply stimulation to this nucleus? If not, is it possible that activation of this nucleus was unrelated to the process in question?

In summary, the strength of the manuscript is in its first phase and that should be highlighted and in the absence of additional neuroanatomical experiments, conclusions regarding direct anatomical connections between the PAG and cuneiform nucleus with respect to flight in mice remain somewhat questionable at best and should be expressed with greater caution.

Authors’ response: In fact, we did carry out retrograde labeling using the traditional retrograde fluorescent protein CTB to show that neurons in dlPAG project to CnF (Figure ?). Because CTB shows relatively limited diffusion from the point of deposition in mouse brain, we can be relatively certain that neurons in dPAG labeled by CTB in this experiment are in fact CnF-projecting dPAG neurons (see Figure 3A-D). Furthermore, we have carried out double labeling for CTB and neuron sub-types in dPAG to show that while CnF-projecting neurons are primarily glutamatergic, there are also GABAergic dPAG neurons that project to CnF (Figure 3D). 

AAV-mediated expression is significantly more diffuse than CTB labeling at the site of deposition (compare Figure 4 to Figure 3). Thus, in our optogenetics experiments where we use viral-mediated retrograde labeling to activate CnF-projecting dPAG neurons we cannot be entirely sure that we are not also activating some dPAG neurons that project to regions close to CnF. However, because light from the optic fibers has limited penetrance capacity and ChR2 expression was limited to dPAG, we can be relatively sure that we are not activating the CnF directly in our retrograde experiments (see fiber placement in Figure 4B). Nevertheless, we now mention the caveat concerning the retrograde labeling of CnF-projecting dPAG neurons in the Discussion. 

Finally, although we did not stimulate neurons in CnF directly, other studies have done so and found that optogenetic activation of glutamatergic neurons there elicits rapid forward movement, consistent with the CnF being a possible conduit for flight behavior (Caggiano 2018). The same study confirmed the existence of direct projections from dPAG to CnF using rabies based mono-synaptically restricted retrograde trans-synaptic circuit tracing. Whether CnF ChR2-evoked locomotion is goal-directly, however, was not tested and we agree that identifying the inputs that confer goal-directed vs non-goal-directed high speed locomotion downstream of dPAG would be an interesting subject for a follow-up study. 

Rebuttal References

Tovote, Philip, Maria Soledad Esposito, Paolo Botta, Fabrice Chaudun, Jonathan P. Fadok, Milica Markovic, Steffen B. E. Wolff, et al. 2016. “Midbrain Circuits for Defensive Behaviour.”

Nature 534 (7606): 206–12.

Blanchard, D. Caroline, and Robert J. Blanchard. 2008. “Chapter 2.4 Defensive Behaviors, Fear, and Anxiety.” In Handbook of Behavioral Neuroscience, edited by Robert J. Blanchard, D. Caroline Blanchard, Guy Griebel, and David Nutt, 17:63–79. Elsevier.

Blanchard, R. J., M. A. Hebert, P. F. Ferrari, P. Ferrari, P. Palanza, R. Figueira, D. C. Blanchard, and S. Parmigiani. 1998. “Defensive Behaviors in Wild and Laboratory (Swiss) Mice: The Mouse Defense Test Battery.” Physiology & Behavior 65 (2): 201–9.

Bolles, R. C. 1970. “Species-Specific Defense Reactions and Avoidance Learning.” Psychological Review. http://psycnet.apa.org/journals/rev/77/1/32/.

Keay, K. A., and R. Bandler. 2001. “Parallel Circuits Mediating Distinct Emotional Coping

Reactions to Different Types of Stress.” Neuroscience and Biobehavioral Reviews 25 (7-8):

669–78.

Caggiano, V., R. Leiras, H. Goñi-Erro, D. Masini, C. Bellardita, J. Bouvier, V. Caldeira, G. Fisone, and O. Kiehn. 2018. “Midbrain Circuits That Set Locomotor Speed and Gait Selection.” Nature 553 (7689): 455–60.

---

## [Editor Report · Decision Letter 1]

24 Jan 2023

Induction of flight via midbrain projections to the cuneiform nucleus

PONE-D-22-23069R1

Dear Dr. Gross,

We’re pleased to inform you that your manuscript has been judged scientifically suitable for publication and will be formally accepted for publication once it meets all outstanding technical requirements.

Kind regards,

Allan Siegel

Academic Editor

PLOS ONE

Additional Editor Comments (optional):

The authors have clearly answered all the questions raised by the reviewer most appropriately and accordingly, it is my opinion that the manuscript should be published. One suggestion to the authors that they can decide or not to decide to follow; namely, since the cuneiform nucleus was a major focus of this paper, it would seem reasonable for the authors to consider a more detailed speculation on the possible role of this structure in the integration of motor and autonomic components of the flight response. Such an addition might strengthen the overall manuscript.
---

## [Editor Report · Acceptance letter]

7 Feb 2023

PONE-D-22-23069R1 

Induction of flight via midbrain projections to the cuneiform nucleus 

Dear Dr. Gross:

I'm pleased to inform you that your manuscript has been deemed suitable for publication in PLOS ONE. Congratulations! Your manuscript is now with our production department. 

Kind regards, 

on behalf of

Dr Allan Siegel 

Academic Editor

PLOS ONE